# Role of Melatonin to Enhance Phytoremediation Capacity

**Marino B. Arnao \***  **and Josefa Hernández-Ruiz**

Department of Plant Biology (Plant Physiology), Faculty of Biology, University of Murcia, 30100-Murcia, Spain; jhruiz@um.es
* Correspondence: marino@um.es

**Abstract:** Phytoremediation is a green technology that aims to take up pollutants from soil or water. Metals are one of the targets of these techniques due to their high toxicity in biological systems, including plants and animals. Their elimination or, at least, decrease will help keep them from being incorporated in the trophic chain and thus reaching animal and human food. The metal removal efficiency of plants is closely related to their growth rate, tolerance, and their adaptability to different environments. Melatonin (*N*-acetyl-5-methoxytryptamine) is a ubiquitous molecule present in animals, plants, fungi, and bacteria. In plants, it plays an important role related to antioxidant activity, but also as an important redox network regulator. Thus, melatonin has been defined as a biostimulator of plant growth, especially under environmental stress conditions, whether abiotic (water deficit and waterlogging, extreme temperature, UV radiation, salinity, alkalinity, specific mineral deficit/excess, metals and other toxic compounds, etc.) or biotic (bacteria, fungi, and viruses). Exogenous melatonin treated plants have been seen to have a high tolerance to stressors, minimizing possible harmful effects through the control of reactive oxygen species (ROS) levels and activating antioxidative responses. Furthermore, important gene expression changes in stress specific transcription factors have been demonstrated. Melatonin is capable of mobilizing toxic metals, through phytochelatins, transporting this, while sequestration adds to the biostimulator effect of melatonin on plants, improving plant tolerance against toxic pollutants. Furthermore, melatonin improves the uptake of nitrogen (N), phosphorus (P), and sulfur (S) in stress situations, enhancing cell metabolism. In light of the above, the application of melatonin seems to be a useful option for clearing toxic pollutants from the environment by improving phytoremediation. Interestingly, a variety of stressors induce melatonin biosynthesis in plants, and the study of this endogenous response in hyperaccumulator plants may be even more interesting as a natural response of the phytoremediation of diverse plants.

**Keywords:** abiotic stress; biostimulators; cadmium; cobalt; copper; heavy metals; lead; nickel; plant growth promoters; zinc

## 1. Introduction

Phytoremediation may be described as a technology that uses the ability of certain plants, known as hyperaccumulators, to take up pollutants from soil or water. Phytoremediation is of particular interest because it is a promising green technology, low cost, and eco-friendly [1]. However, hyperaccumulator plants are rare, and generally, their slow growth and small biomass limits their efficiency for phytoremediation purposes. The hyperaccumulator plants used for phytoremediation mainly accumulate metals in the shoot rather than the root [2]. Compared with non-accumulator plants, hyperaccumulating plants can concentrate 100 to 1000 times more heavy metals such as copper (Cu), zinc (Zn), cobalt (Co), manganese (Mn), nickel (Ni), and lead (Pb) [3]. Each plant species acts in a specific way in phytoremediation, taking up heavy metals by many mechanisms, including

accumulation, exclusion, translocation, osmoregulation, and distribution. Of these, the most common is via the accumulation, translocation, and concentration of heavy metals in the aerial parts [4].

Various processes have been used as phytoremediation techniques, including phytoextraction, phytodegradation, rhizofiltration, phytostabilization, phytovolatilization, phytodesalination, and phytofiltration [5]. Phytoremediation can be carried out on site, thus reducing exposure risks for cleanup personnel or secondary contamination during transport. However, the physical, chemical, and biological properties of mine tailings (or other contaminated soils) may limit plant growth and their subsequent use in agriculture [6]. Three major phytoremediation techniques can be distinguished, depending on different plant properties: phytoextraction, phytostabilization, and phytovolatilization [7]. In the first method, plants are used to take up contaminants or metals via their roots. This type of extraction involves the accumulation or hyperaccumulation of metals in the above ground plant biomass. Contaminants or metals are then stored in the harvestable parts of plants and disposed of as hazardous waste or incinerated for metal recovery. In the case of phytostabilization, plants are used as a vegetation "cap" to not only reduce the mobility and bioavailability of contaminants in the natural environment, but also the availability for entry into the human food chain. The plant canopy serves to reduce air dispersion, while plant roots prevent water erosion, immobilize heavy metals, and prevent leaching. Thus, phytostabilization is a promising technique for the long term stabilization of tailings by creating a vegetation cap. Finally, phytovolatilization involves plants taking up pollutants (including organic contaminants) together with water and releasing them into the atmosphere through the stomata; some of these pollutants pass through the plants and reach the leaves and thereby evaporate into the atmosphere [8,9].

In constructed wetlands, the interaction between water/soil, plants, and microorganisms occurs through chemical, physical, and biological processes. A wide range of wastewaters such as municipal, industrial, agricultural, and storm waters can be remediated in constructed wetlands. The efficiency of heavy metal uptake by the plants they contain has been demonstrated [10]. The rate of metal removal by plants varies widely and is related to plant growth rate, plant species, and the concentration of heavy metals in the wastewater [11]. In wetlands constructed for phytoremediation or wastewater treatment, the residence time of metals in plants and the potential release of metals closely depend on the distribution of metals within the plants. Artificial floating islands, another type of constructed wetlands, are a soil-less planting structure constructed with floating mats, floating aquatic plants, sediment rooted emergent wetland plants, and related ecological communities. In pilot studies, this system was seen to improve the quality of polluted waters by removing organic matter, suspended solids, nutrients, and metals [12].

For the above described systems, aquatic plant species are of interest for use in phytoremediation processes because they can accumulate more than 1450-fold the concentration of the heavy elements in water [13]. In free-floating macrophytes, the entire plant body is above water except the roots, while in submerged macrophytes, the whole plant body remains submerged in water. For their part, emergent macrophytes are plants rooted in soil, but which emerge to reach significant heights above the water. A wide range of wetland plant species, such as water hyacinths *Eichhornia* spp., *Salvinia* spp., water lettuce (*Pistia stratiotes*), giant duckweed, duckweed (*Lemna minor*), and *Azolla* spp., submerged species such as *Potamogeton* spp. and *Myriophyllum* spp., and emergent species like *Typha* spp., *Scirpus* spp., *Limnocharis flava, Spartina* spp., *Cyperus* spp., and *Phragmites* spp. have shown their potential for removing metals from various type of wastewaters [14,15]. An extensive web-list of plant hyperaccumulators, classified by the metals they absorb, can be consulted [16].

Selecting the appropriate plant species is one of the most important considerations in the phytoremediation process. The plant species chosen should be capable of tolerating high metal levels and extreme soil conditions, such as high acidity, salinity, or alkalinity [17]. In addition, plants for revegetation should have other favorable attributes such as dense rooting systems, relatively fast growth rates, and high biomass production [7]. Furthermore, in semiarid mining regions, plant species

should also be able to adapt to drought. Metal tolerant native plants are usually selected because they demonstrate tolerance to local environmental conditions and usually grow well and proliferate [18].

Suitable plants for phytoremediation can be divided into two groups based on their function: metal hyperaccumulators and biomass producers [19,20]. Metal hyperaccumulators are plants that exhibit higher levels of metal ion absorption in their tissues, but usually do not produce a high amount of biomass and have a slow growth rate. *Brassicaceae* spp. are known to have exceptionally high metal accumulating capacities [21]. Some authors have reported that *Thlaspi* species (typical hyperaccumulators) can accumulate over 1% Zn, 0.1% Ni, and 0.1% Cd in dry tissues. In addition, other heavy metals, including arsenic (As), Cu, Co, Mn, and Pb, can potentially be hyperaccumulated from mine tailings [19,20]. Biomass producers include plants that have a high biomass production and growth rate, but a relatively low metal uptake capacity, such as *Brassica juncea* (Indian mustard) [21].

Nutrient shortages are one of the main limitations of plants used for phytoremediation in mining areas, so it is generally necessary to amend the soils. Other biotechniques include: (i) enrichment with microorganisms, which can play an important role in solubilizing minerals such as P and potassium (K), releasing nutrients, and supplying them to plants through in situ bioremediation processes [22]; (ii) the use of metabolites such as organic acids, amino acids, and vitamins, which have also been demonstrated to enhance plant growth [23]; (iii) the addition of plant hormones, siderophores, and some enzymes synthesized by microorganisms, which may help plant growth [24–26]. In general, plants inoculated with plant growth promoting bacteria (PGPB) accumulate greater amounts of heavy metals than non-inoculated plants [27].

## 2. How Can Melatonin Contribute in a More Efficient Phytoremediation?

Decarboxylation of aromatic amino acids by specific decarboxylases leads to the production of starter compounds for the biosynthesis of secondary metabolites involved in stress resilience mechanisms [28]. More in particular, plant tryptophan decarboxylase (TDC) converts tryptophan into tryptamine [28], the precursor of N-acetyl-5-methoxytryptamine, known commonly as melatonin [29]. The protective effect of melatonin against abiotic stress situations in plants has been widely studied. Melatonin acts as an effective free radical scavenger against harmful reactive molecules, both reactive oxygen (ROS) and reactive nitrogen (RNS) species, such as hydroxyl radical, superoxide anion, singlet oxygen, hydrogen peroxide, hypochlorous acid, nitric oxide, peroxynitrite anion, peroxynitrous acid, and lipid peroxyl radicals, among others. The excellent properties of melatonin as an in vivo antioxidant against ROS/RNS, the absence of pro-oxidant effects, and the cascade antioxidant effect of melatonin related compounds have been the objective of a great number of researchers [30]. Melatonin is a more effective antioxidant than vitamin C and E, with a scavenging activity 4–6 fold higher [31]. Furthermore, the amphipathic properties of melatonin permit it to scavenge free radicals in both hydrophilic and lipophilic media [30,32–38]. This direct chemical action of melatonin with ROS and RNS has been referred to as a receptor independent action [39].

In addition to this direct action, melatonin is capable of inducing many changes in gene expression. Melatonin changes the expression of a great number of gene elements in different physiological situations both in plant and animal cells [40–42]. Among the most studied melatonin mediated aspects in plants are the responses to abiotic stress (heat, cold, drought, salinity, alkalinity, heavy metals, and other toxic agents such as herbicides, fungicides, diverse contaminants) and biotic stress (fungi, virus, and bacteria). Furthermore, processes such as foliar senescence, growth and development, germination, rooting induction, flowering, parthenocarpy, fruit set, and fruit ripening were studied. Others aspects, such as photosynthesis and its regulation, primary and secondary metabolism, including osmoregulation, and the regulation of plant hormones (auxin, gibberellins, cytokinins, abscisic acid, ethylene, jasmonates, salicylic acid, polyamines, brassinosteroids, strigolactones) have been also analyzed [41,43–49].

In general, melatonin induces several gene expression changes in plants that result in a biostimulating response [41]. Due to the diversity of its actions, melatonin has been proposed

as a plant master regulator, but also as a new plant hormone since its receptor (PMTR1) has been identified in *Arabidopsis* [41,50]. Melatonin acts as a main regulator of the redox network in plants, controlling directly and indirectly the ROS/RNS levels and the gene expression of many factors through the nitric oxide signaling cascade, among other pathways [51]. As a result, melatonin regulates redox network homeostasis, balancing several ROS and RNS and related key enzyme expressions (NOS-like, NR, RbOHs, ASA-GSH cycle, antioxidant enzymes). It is also one of the main intermediates in many cellular and physiological responses, as is depicted in the integrative model of Figure 1.

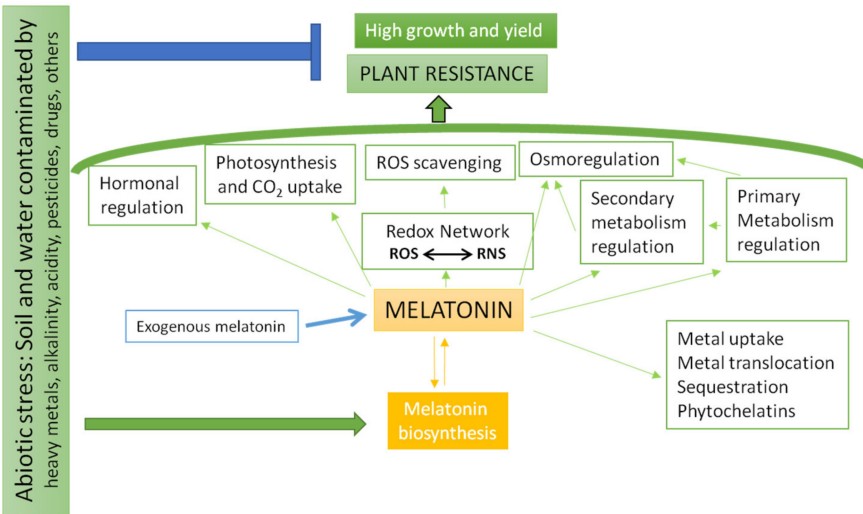

**Figure 1.** Model of the redox network regulated by melatonin and the effect of abiotic stressors.

## 3. Metals as a Severe Abiotic Stress and Effects Induced by Melatonin in Plants

The metal removal efficiency of plants is highly related to their growth rate, tolerance to high levels of metals, and adaptability to different environments. Melatonin is widely distributed among different plants, including hyperaccumulators such as water hyacinth, sunflower, mustard plants, radish, etc. [43], which suggests a probable role in phytoremediation, especially due to its high levels.

Table 1 shows several studies with metals and the effects observed by exogenous melatonin treatment. Furthermore, several studies with transgenic plants overexpressing melatonin biosynthesis genes, which extra-accumulate endogenous melatonin, have been made [44].

Several metals and non-metal elements have been used to study the resistance, tolerance, mobilization, accumulation, and diverse metabolic and physiological changes that occur in plants treated with melatonin compared with control plants. The first report related to phytoremediation and melatonin was published by Tan and co-workers (2007), who observed high contents of melatonin and melatonin by-products in water hyacinth [52]. The capacity of these plants to tolerate high levels of toxic pollutants was proposed. In a related study, the tolerance of pea plants to copper (Cu) significantly improved after supplementation with melatonin [53].

Cadmium (Cd) has been the most widely assayed heavy metal in melatonin treated plants of several different species (Table 1). In general, melatonin treatment increases Cd tolerance, plant growth, and photosynthesis efficiency compared with non-treated plants. Furthermore, their water content and ion homeostasis were improved. Antioxidant enzymes increased and the ASA-GSH cycle improved, while ROS and MDA contents were reduced, optimizing the redox balance. Cd transport increased with the phytochelatin content in alfalfa and tomato, where the Cd content tended to accumulate in different locations (roots, stems, leaves, shoots). When the effects of melatonin on growth and Cd uptake in *Malachium aquaticum* (Cd accumulator) and *Galinsoga parviflora* (hyperaccumulator) were studied, melatonin application significantly increased the Cd content in both plants in a concentration dependent form, suggesting melatonin could improve phytoremediation [54]. In *Cyphomandra betacea*, a South American fruit tree, 50 μM melatonin increased the Cd contents in stems, leaves, and shoots in

a soil cultivation experiment. Low levels of melatonin were seen to promote the growth of *C. betacea* seedlings and their Cd accumulation capacity [55]. Melatonin treated plants, in a combination of Cd with selenium (Se) or Zn, also showed increased plant growth and higher antioxidative defenses than control plants. In an interesting experiment with tomato plants that over-produced melatonin, pre-treatment with different forms of Se significantly induced the biosynthesis of melatonin and its precursors (tryptophan, tryptamine, and serotonin) [56].

**Table 1.** Studies of responses to melatonin treatments in the presence of several metals (or others) by different plant species.

| Stress Type | Plant Species | Melatonin Treatment (µM) | Effects Observed | Reference |
|---|---|---|---|---|
| Cd | Alfalfa | 10–200 | ↑ tolerance, growth, Cd transporters, ↓ Cd roots, ROS | [57] |
| Cd | Rice | MOE * | ↑ tolerance, growth, photosynthesis, redox balance, panicle number, grain yield | [58,59] |
| Cd | Tomato | 25–500 | ↑ Cd tolerance, phytochelatins, ATPase activity | [60] |
| Cd | Tomato | MOE | ↑ Cd tolerance, heat-shock factor A1a, induced by melatonin | [61] |
| Cd | Tomato | 25 | ↑ Cd tolerance, melatonin biosynthesis, ↓ Cd leaf | [62] |
| Cd | Tomato | 100 | ↑ Cd tolerance, melatonin biosynthesis, S uptake, S assimilation, antioxidant enzymes, PCs, GSH | [63] |
| Cd | Wheat | 100 | ↑ tolerance, antioxidant enzymes, ASA, GSH, ↓ ROS | [64] |
| Cd | Wheat | 50–100 | ↑ tolerance, plant growth, Chls, PSII maximum efficiency, RWC, $K^+$, $Ca^{2+}$, ↓ ROS, Cd, MDA, NO | [65] |
| Cd | *Malachium aquaticum* and *Galinsoga parviflora* | 100–200 | ↑ tolerance, biomass, Chls, antioxidant enzymes, Cd shoots concentration dependent | [54] |
| Cd | *Cyphomandra betacea* | 50–150 | ↑ plant growth, Cd leaves, shoots, stems, antioxidant enzymes | [55] |
|  | *Perilla frutescens* | 100–200 | ↑ root and shoot biomass, Chls, antioxidant enzymes, soluble protein, Cd root and shoot | [66] |
| Cd/Se | Tomato | MOE | ↑ growth, photosynthesis, electrolyte leakage, phytochelatins, GSH, ↓ ROS, Cd leaf | [56] |
| Cd/Zn | Valerian and Lemon balm | 1000 | ↑ tolerance, plant growth, antioxidant enzymes | [67] |
| Cu | Red cabbage | 1–100 | ↑ germination, growth, ↓ membrane peroxidation | [68] |
| Cu | Cucumber | 0.01 | ↑ tolerance, growth, Cu-sequestration, TCA, ATP, GSH, ↓ ROS | [69] |
| Cu | Pea | 5 | ↑ plant survival | [53] |
| Zn | Wheat | 1000 | ↑ tolerance, Chls, photosynthesis, Rubisco, ATPase | [70] |
| Al | Soybean | 0.1–1 | ↑ tolerance, root growth, antioxidant enzymes, osmoregulation, ↓ ROS | [71] |
| Al | *Arabidopsis* | 1–10 | ↑ tolerance, root growth, cell division | [72] |
| Pb | Maize | 50–100 | ↑ tolerance, growth, photosynthesis, Chls, RWC, K, Ca levels, ↓ ROS, MDA | [73] |
| Pb | Bermudagrass | 20–100 | ↑ tolerance, biomass, Chls, RWC, ASA, GSH, antioxidant enzymes, ↓ ROS, lipid peroxidation | [74] |
| V | Watermelon | 0.1 | ↑ tolerance, growth, photosynthesis, antioxidant enzymes, ↓ V level, V transport, ROS, MDA | [75] |
| Boron | Pepper | 1 | ↑↑ tolerance, growth, photosynthesis, antioxidant enzymes, carotenoids, ↓ B in leaf and fruit, toxicity, ROS, MDA | [76] |
| Boron | Spinach | 100–300 | ↑ tolerance, growth, photosynthesis, RWC, $CO_2$ uptake, sugars, carotenoids, redox balance, ↓ ROS, MDA | [77] |
| Fluoride | Pigeon pea | 100 | ↑ tolerance, growth, antioxidant capacity, protein, proline, ASA, GSH, antioxidant enzymes, genomic template stability, ↓ ROS, cell death, lipid peroxidation, lipase activity, DNA polymorphism | [78] |
| Alkalinity | Apple | 5 | ↑ tolerance, root system, redox balance, polyamines | [79] |
| Alkalinity | Tomato | 0.25–1 | ↑ seedling growth, photosynthesis, ion homeostasis, $Na^+$ detoxification, dehydration resistance, ROS homeostasis, *DREB1α* and *IAA3* transcription factors | [80,81] |

**Table 1.** *Cont.*

| Stress Type | Plant Species | Melatonin Treatment (µM) | Effects Observed | Reference |
|---|---|---|---|---|
| Acid rain | Tomato | 100 | ↑ tolerance, growth, chloroplast integrity, photosynthesis, antioxidant enzymes, ↓ ROS, MDA | [82] |
| Salinity, Fe-low | Pepper | 100 | ↑ growth, Chls, photosynthesis, fruit yield, Fe, K uptake, antioxidant enzymes | [83] |
| Fe-low | *Arabidopsis* | 5 | ↑ melatonin, Fe shoots and roots, Fe mobilization, NO, polyamines, ↓ chlorosis, Fe root cell walls, ROS | [84] |
| S-low | Tomato | 100 | ↑ S uptake, assimilation, transport and metabolism, peroxiredoxins, redox homeostasis, ↓ ROS, DNA damage | [85] |
| N-low | Wheat | 1 | ↑ N and nitrate, N absorption, N metabolism, growth, yield, in shoots and roots | [86] |
| N-excess | Cucumber | 100 | ↑ tolerance, growth, NPK balance, Ca, ↓ damage, nitrate, ammonium | [87] |
| Cinnamic acid | Cucumber | 100 | ↑ tolerance, growth, water and nutrient balance, hormonal balance | [88] |
| Butafenacil | Rice | MOE | ↑ herbicide tolerance, Chls, antioxidant enzymes, ↓ ROS, MDA | [89] |
| Fluopicolide | Potato | 1–10 | ↑ fungicide tolerance, ↓ ROS, potato late blight, mycelial growth of *P. infestans* | [90] |
| Paraquat | Pea | 50–200 | ↑ Chls, porphyrin synthesis pathway, ↓ herbicide damage, Chl breakdown | [91] |
| Carbendazim | Tomato | 0.5/MOE | ↑ fungicide tolerance, antioxidant enzymes, ASA-GSH cycle, ↓ ROS, MDA | [92] |

↑, Increased content or increased action. ↓, Decreased content or decreased action. * MOE, melatonin biosynthesis enzymes overexpressed in plants.



Selenocysteine had the most marked effect on melatonin biosynthesis. Se treatment increased the levels of glutathione (GSH) and phytochelatins, as well as the expression of GSH and phytochelating biosynthetic genes in non-silenced plants, but the effects of Se were lessened in *TDC* silenced plants (low melatonin content) under Cd stress. Furthermore, Se and melatonin supplements significantly increased plant Cd tolerance, optimizing plant growth parameters. Thus, exogenous selenocysteine could ameliorate Cd phytotoxicity, but a basal level of endogenous melatonin is required for Se conferred Cd tolerance, which might enhance the detoxification of Cd (Table 1) [56].

Copper (Cu) salts have also been used to study the protective and biostimulatory effect of melatonin (Table 1). In addition to its known effect on the germination of seeds in the presence of Cu, which was one of the first roles of melatonin discovered [68,93], low levels of melatonin provide greater tolerance of the presence of Cu in cucumber plants. Melatonin induces metabolic activity, especially glycolysis and the pentose phosphate pathway, to generate more ATP. Melatonin treatment broadly altered gene expression under Cu stress, increasing the levels of GSH and phytochelatin to chelate excess Cu and promoting cell wall trapping, retaining more Cu in the cell wall and vacuole [69]. Furthermore, some experiments have been made using zinc, aluminum, and lead in wheat, soybean, *Arabidopsis*, and maize (Table 1).

Vanadium (V) adversely affects plant growth through drastic changes in cellular metabolism, including gene expression and ROS production. Higher levels of V affect root growth and the formation of lateral roots and provokes leaf chlorosis. Vanadium is a chemical analogue of phosphorus (P) and alters the P absorption capacity of plants [94,95]. In a recent study, melatonin pretreatment lowered leaf and stem V concentrations by reducing V transport from root to shoot in watermelon (Table 1). In V treated plants, melatonin also renewed plant growth and increased photosynthesis efficiency; the antioxidant enzyme pool was also improved. Taken together, this evidence underlines the protective role of melatonin in increasing V tolerance [75].

Boron (B) is an essential micronutrient for normal plant growth, but high concentrations are toxic for plants. Boron has also been assayed in the presence of melatonin (Table 1). In pepper plants whose roots were exposed to a high B concentration (100 μM), melatonin treatment restored plant growth and photosynthesis compared with control plants. Plants treated with melatonin displayed no visible B toxicity symptoms, and leaves and fruits showed moderate B accumulation and high carbohydrate, carotenoid, and flavonoid contents. The authors took this as a demonstration of the clear protective activity of melatonin in reducing B absorption, suggesting its physiological relevance in B homeostasis [76]. Similar results and conclusions have been obtained in spinach plants exposed to high B concentrations (Table 1).

Higher concentrations of fluoride ions ($F^-$) in the soil and irrigation water can disturb both the physiological and biochemical processes of plants [96]. Melatonin acts as an ROS scavenger, improving many biochemical parameters in pigeon pea (*Cajanus cajan*) (Table 1), helping in diminishing $F^-$ toxicity [78].

Although soil alkalization is often associated with soil salinity, the former is considered much more hazardous to plants. This condition is generally linked with high pH stress and sodium toxicity caused by an excess of $Na_2CO_3$ and $NaHCO_3$ in the soil, as well as osmotic stress. The comprehensive stress caused by alkaline soils directly affects physiological homeostasis at the cellular and whole-plant levels [97]. In this context, melatonin has been seen to protect apple and tomato plants against alkalinization (Table 1). In the case of apple, exogenous melatonin enhanced tolerance to alkaline stress by regulating the biosynthesis of polyamines, while in tomato plants, a strong resistance to dehydration, ROS homeostasis, and $Na^+$ detoxification has been described [80,81]. In tomato also, melatonin treatment increased simulated acid rain (SAR) stress tolerance by repairing the grana lamella of the chloroplast, improving photosynthesis and antioxidant enzyme activities compared with the reactions recorded in SAR stressed plants without melatonin. Such positive effects of melatonin are concentration dependent [82].

The positive role of melatonin against soil mineral deficiency has been described in several works (Table 1), resulting in an improvement in chemical element deficiency tolerance. For example, Fe deficiency induced chlorosis in seedlings of *Arabidopsis thaliana* was alleviated by melatonin. Exogenous melatonin significantly increased the soluble Fe content of shoots and roots and decreased the levels of root cell wall Fe bound to pectin and hemicellulose, remobilizing cell wall Fe and alleviating Fe deficiency induced chlorosis. Furthermore, Fe deficiency quickly induced melatonin biosynthesis in *Arabidopsis* plants, acting synergistically with exogenous treatments. In mutant plants deficient in polyamine and nitric oxide (NO) biosynthesis, the protective role of melatonin was not observed, indicating that the process is dependent on the polyamine induced NO production under Fe deficient conditions [84]. In a similar work, tomato seedlings grown in an S deficient medium suffered serious growth inhibition as a result of a reduced chlorophyll content, photosynthesis, and biomass accumulation; it also led to cell structural alterations and DNA damage. Melatonin supplementation of S deprived plants resulted in a significant diminution in ROS content, alleviating all the described symptoms. Melatonin promoted S uptake and assimilation by regulating the expression of genes encoding enzymes involved in S transport and metabolism, supporting a role for melatonin as a molecule that improves primary metabolism and redox homeostasis [85].

In winter wheat grown in an N deficient medium, the application of melatonin in hydroponic solution significantly improved seedling growth under both N sufficient and deficient conditions, but the effect of melatonin in promoting seedling growth was particularly evident in the N deficient conditions. Higher N contents and nitrate levels in shoot under N deficient conditions appeared and also maintained higher nitrate nitrogen levels in roots. Furthermore, nitrate reductase and glutamine synthetase activities were higher in melatonin treated plants under N deficiency conditions. In conclusion, melatonin is involved in promoting N uptake and assimilation through upregulating the activities of N uptake and metabolic related enzymes and, ultimately, promotes plant growth and yield, especially under N deficient conditions [86]. Melatonin also seems to play an important role in situations contrary to those described above. Excess nitrogen is generally applied so that adequate levels will be maintained in the rhizosphere. This abusive use, besides being a serious problem since it results in the contamination of aquifers, provokes disruption in the balance of elements and alters the assimilation of calcium and magnesium, affecting the susceptibility to disease. In general, nitrate accumulation leads to increased proline concentrations, severe oxidative damage, nitrogen metabolic disorders, the inhibition of photosynthesis, and a substantial decrease in biomass. The application of melatonin significantly improved the growth of cucumber plants and reduced their susceptibility to damage when grown in high nitrate levels. Although excess nitrate led to an increase in the concentrations of N, K, and Ca, accompanied by a decrease in P and Mg levels, exogenous melatonin generally had the opposite effect, except for a further rise in Ca concentrations (Table 1). Pretreatment also significantly reduced the accumulations of nitrate (both N and ammonium) and enhanced the activities of the enzymes involved in nitrogen metabolism, thereby alleviating the inhibitory effect on growth normally associated with nitrate stress [87].

Another role of melatonin is as a biostimulator and alleviator of toxicity from herbicides and other chemicals (Table 1). For example, melatonin significantly attenuated potato late blight by inhibiting mycelial growth, changing the cell ultrastructure, and reducing the stress tolerance of the high pathogenic fungus *Phytophthora infestans*. Of particular note are the synergistic anti-fungal effects of melatonin and the fungicide Infinito (fluopicolide, Bayer®) on *P. infestans*, suggesting that melatonin could reduce the dose levels and enhance the efficacy of the fungicide against potato late blight [90]. The same synergistic action of melatonin and ethylicin (a bio-oomyceticide fungicide) has been described [98]. A similar beneficial effect of melatonin treatments was observed with other fungicides and herbicides (Table 1).

## 4. Abiotic Stressors Induce a Melatonin Burst that Activates Anti-Stress Responses

Endogenous melatonin levels change with environmental conditions of plant growth. Melatonin is accumulated as a protective molecule in response to different environmental abiotic stressors, such as water deficit and waterlogging, cold and heat, UV radiation, soil heavy chemicals and related, among others [41,48,99]. Table 2 shows some studies on the increase of endogenous melatonin levels by the presence of different abiotic stressors. Thus, the expression of the biosynthesis enzyme transcripts (TDC, SNAT, ASMT, and COMT genes) occurs in stress situations, producing a burst in the levels of endogenous melatonin. The global influence of environmental factors on the melatonin levels of plant organs was clearly demonstrated in barley, tomato, and lupin plants by Arnao and co-workers [99–101]. This effect was previously suggested in water hyacinth plants [52], and later corroborated in grape berry skin [102] and cherry fruits [103]. Salinity, cold, drought, and heavy metals have been the abiotic environmental agents most frequently studied as inducers of melatonin biosynthesis in plants, although attack by pathogens also induces the biosynthesis of melatonin (Table 2). This interesting response of stressed tissues clearly induces tolerance, fortifying the redox network against ROS and RNS and upregulating the expression of stress specific response genes. All this relieves the inhibitory processes due to stress and reinforces plant growth and critical processes such as photosynthesis, water economy, metabolism, etc. [41,51,104,105]. The remediation efficiency is directly correlated to higher biomass and an improved tolerance of plants to toxic pollutants. Studies made with melatonin suggested that the stimulatory effects of melatonin on biomass and antioxidative defense machinery are reinforced by a strong primary and secondary metabolism and also by plant hormone stress responses. Melatonin acts as a biostimulator and/or protector of photosynthesis and the stomatal apparatus, upregulating many elements of photosystems, thylakoid electron transporters, and ATP-ase genes. Melatonin also optimizes stomata functionality (e.g., by causing increased stomata opening) in adverse conditions through the regulation of guard cell anion channel proteins and dehydrins, all of which increase $CO_2$ availability. In the Calvin cycle, melatonin regulates the expression of RuBisCO elements, glyceraldehyde-3-phosphate dehydrogenases, and interconversion carbohydrate enzymes. In addition, elements of the ASC-GSH cycle, TCA cycle, and myo-inositol and fatty acid biosynthesis pathways are also regulated by melatonin. As regards osmoregulation, higher levels of proline, carbohydrates (glucose, maltose, fructose, sucrose, and trehalose), and a multitude of amino acids and organic acids in melatonin treated plants had beneficial effects under abiotic stress conditions. Thus, relevant changes in carbohydrate, lipid, amino acid, nitrogen, phosphorus, and sulfur metabolism indicate the beneficial physiological processes that occur in melatonin treated plants during abiotic stress [86,87,106–109]. Melatonin is also involved in secondary metabolism, where it induces anthocyanin biosynthesis and flavonoids [110,111] and also regulates steps in the carotenoid biosynthesis [76,112]. Finally, melatonin regulates the expression of multiple elements (enzymes, receptors, and transcription factors) in the biosynthesis, catabolism, and signaling of auxin, gibberellins, cytokinins, ABA, ethylene, jasmonic acid, SA, brassinosteroid, strigolactones, and polyamines [41,46,48] (Figure 1).

**Table 2.** Effects of abiotic stressors on the endogenous melatonin levels.

| Plant Species | Abiotic Stressor | Increased Level of Melatonin vs. Control | Reference |
|---|---|---|---|
| Alfalfa | Waterlogging | 2–4.5-fold | [113] |
| *Arabidopsis* | Cold | 2-fold | [114] |
| | Heat | 2–5-fold | [115] |
| | NaCl, drought, cold | 3–6-fold | [116] |
| | Fe deficiency | 6-fold | [84] |
| | Drought | 4-fold * | [117] |
| Barley | Zn, NaCl, $H_2O_2$ | 6-fold | [100] |
| Barley | Drought, cold | 2-fold | [118] |
| Bermudagrass | NaCl, drought, cold | 2–3-fold | [106,119] |
| Cassava | Bacterial blight | 1.2–4-fold * | [120] |
| Cherry | Field growth conditions | 10-fold | [103] |
| Grape | Field growth conditions | 15-fold | [102] |
| Lupin | Zn, NaCl, $H_2O_2$ Cold, drought | 1.5–12-fold | [99] |
| *Malus* | Drought | 1.5–6-fold * | [121] |
| Rice | Cd | 6-fold | [122] |
| | Cold, salt, drought, pathogen | 1.5–4.5-fold * | [123] |
| Ryegrass | Darkness | 2-fold | [124] |
| Sunflower | NaCl | 2–6-fold | [125] |
| Tomato | Field growth conditions | 10-fold | [101] |
| | Cd | 1.6–4-fold | [62] |
| | Cd | 2-fold | [60] |
| | Cd | 2–15-fold * | [63] |
| | High temperature | 2–15-fold * | [126] |
| *Vitis* | NaCl | 5.5-fold | [127] |
| | Osmotic | 1.5-fold | |
| Water hyacinth | Field growth conditions | 2-fold | [52] |
| Watermelon | V | 4-fold * | [75] |

* Number of increments of one or more transcripts of melatonin biosynthesis enzymes due to stressor presence.

## 5. Conclusions and Expectations

The application of melatonin on plants seems to be a useful option for cleaning toxic pollutants from the environment by improving phytoremediation processes. In this capacity, three aspects should be taken into account: (i) the inhibitory effect of metals or toxic agents on growth and other basic functions such as metabolism and photosynthesis, (ii) the ability of cells to mobilize, absorb, and sequester the said agents, and (iii) other adverse elements that usually accompany the presence of contaminants such as drought, salinity, extreme temperatures, etc. The beneficial effects of melatonin described above were seen to cover all three cases. The potential of melatonin to mobilize toxic metals, through phytochelatins, their transport, and sequestration adds to the general biostimulatory effect of melatonin on plants, resulting in a high degree of plant tolerance against toxic substances (Figure 2). Furthermore, the improvement in the absorption and metabolism of elements such as N, P, and S helps the process. Although data are still limited, it seems that the presence of several stressors (e.g., metals and drought) synergistically induces the response of melatonin biosynthesis, which reinforces the overall response of the system. Beyond the use of transgenic plants that overaccumulate melatonin, the application of exogenous melatonin or the induction of its biosynthesis through environmental elicitors can be excellent strategies for phytoremediation purposes. There are clear benefits to be had from further studying the applicability of melatonin for phytoremediation purposes. As regards plant species, more research into the biochemical and physiological aspects of melatonin in hyperaccumulator plants is indispensable, furthering our knowledge of the synergistic effect of abiotic stressors on endogenous melatonin levels and its phytoremediation capacity. Furthermore,

the modes of application of exogenous melatonin through the roots and/or through the leaves should be studied, although its amphipathic nature already means that we know that it is well absorbed at both sites (rhizosphere and leaves) without the need for adjuvants and that it is easily transported throughout the plant. Sufficient data are available to suggest the potential of melatonin to improve phytoremediation, but the last decisive step needs to be taken: testing in real field situations.

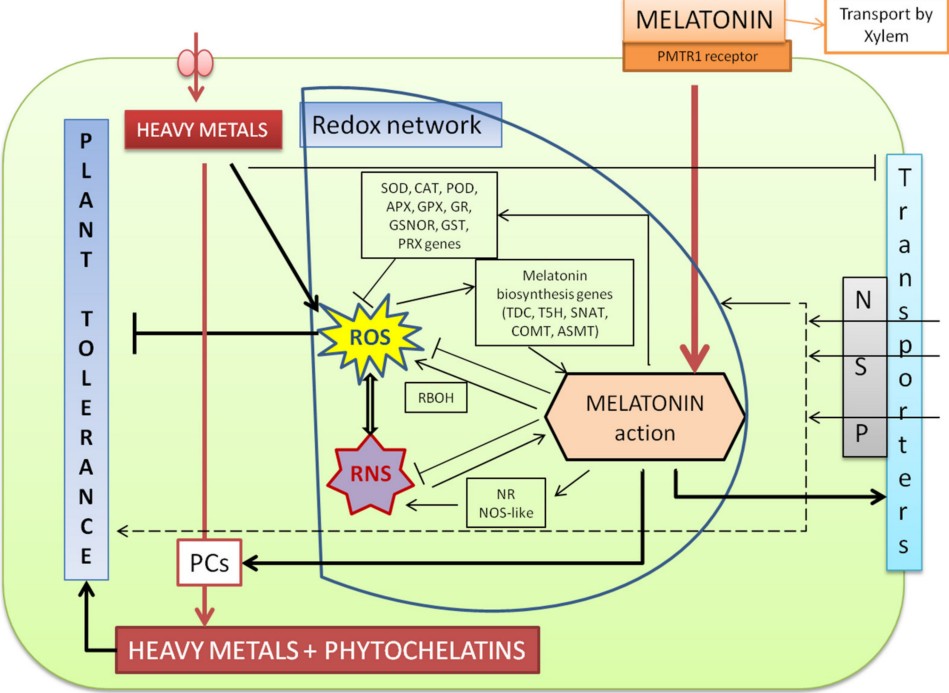

**Figure 2.** Integrated model of melatonin, the redox network, and phytochelatin (PCs) action focused on plant tolerance against heavy metal contaminants in plant cells.

**Author Contributions:** The manuscript was conceived by M.B.A. and written by M.B.A. and J.H.-R.

**Funding:** No external financial support available for this review.

**Conflicts of Interest:** The authors declare no conflicts of interest.

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
