# Peer review of "Role of Melatonin to Enhance Phytoremediation Capacity"

_applsci, doi:10.3390/app9245293_

Round 1

Reviewer 1 Report

MS: applsci-647021

After careful review of the manuscript entitled “Role of melatonin to enhance phytoremediation capacity of plants” by Arnao and Ruiz (Manuscript ID: applsci-647021), this reviewer suggests its publication in Applied Sciences after minor revisions.

This paper is a comprehensive update on past and recent discoveries of melatonin roles in planta applied to improve phytoremediation.

A drawback of this ms is that it lacks of clarity of presentation in some points. English language needs improvement, also checking typing errors. Accordingly, this reviewer recommends the authors to perform a revision of their paper. The rest of comments and suggestions is in the following points.

TITLE

This reviewer suggests to modify the present title to “Role of melatonin to enhance phytoremediation capacity”, note that “plant” is already intrinsic in “phyto…”

ABSTRACT

L8-10: the second part of this sentence is not clear, check English and rewrite: “Metals are one of the targets of these techniques due to their high toxicity in biological systems including plants and animals, which way contribute to them entering the trophic chain and hence the food destined for human consumption.”

L22: check English for “it”

L24-25: remove “and help to the process”

L27: check English and insert “a” before variety

KW

L31-32: insert “nickel”, “cobalt”, “manganese”, “lead”, and “arsenic” as reported in L42 and L13

Pag. 1

L37: consider to use “low cost” instead of “inexpensive”

L40: remove comma before “mainly”

Pag. 2

L8: remove “constructed wetlands” before “to uptake”

L9-11: remove the sentence “In such constructed wetlands…”

L24: change “Another type” to “Also”, remove comma before “are”

L26-29: change “sp.” to “spp.” for the plural species

L29: use plural for “type”

L30: insert comma before “classified”

L31: insert reference or web site

L35-49: move this part to L5, after “[5].”

L50: if the first sentence is a subtitle, then remove it, if not is then complete it.

Pag. 3

L11: change “species” to “spp.”

L24: use full description before acronym PGPB

L28: Here, this reviewer would suggest an essential integration to text and two recent references on the first biosynthetic step of melatonin in plant, consequently renumber the references:

“Decarboxylation of aromatic amino acids by specific decarboxylases leads to the production of starter compounds for the biosynthesis of secondary metabolites involved in stress resilience mechanisms [27]. More in particular, plant tryptophan decarboxylase (TDC) converts tryptophan into tryptamine[27], precursor of N-acetyl-5-methoxytryptamine known commonly as melatonin [28].”

Molecules 2017, 22, 272; doi:10.3390/molecules22020272 Molecules 2018, 23, 1164; DOI: 10.3390/molecules23051164

L38-39: insert the appropriate reference

L48: change “also the” to “and”, remove “of key elements”

Pag. 4

L2: insert the appropriate reference

L19: insert “and effects induced by melatonin” after “…in plants”

L25-27: this sentence is not clear, check English and rewrite

L31: insert “(2007)” after “co-workers”

Pag. 5

L2: insert “(Cu)” after “copper”

L5-6: complete the sentence: “Also, water content and ion homeostasis.”

L17-20: insert the appropriate reference

Table 1

In correspondence of Ref. [83], check “Cinmamic acid”

Pag. 8 (not numbered)

L2: insert description of acronym GSH

L19: change “provokes” to “provoke”

L25: change full stop to comma, and “However,” to “but”

L28: remove details of control: (100 μM B treated without melatonin)

Pag. 9 (numbered as 2 of 22)

L6: remove space after “that”

L33: clarify the part “nitrate (N and ammonium)”

L46: change “environmental plant growth conditions” to “environmental conditions of plant growth”

L49-50: check English

Pag. 10 (numbered as 3 of 22)

L1-2 and 3: check repetition of “The global influence of environmental factors on the melatonin levels”

L8: remove “a” and “response” (repetition in L7)

L9,10: change to “relieve” and to “reinforce”

L12: remove “plants”

Pag. 11 (numbered as 4 of 22)

L3: insert “on plants” between “melatonin” and “seems”

L4: remove “phytoremediative” and “decisive”

L9: remove “on plants”, change “seem” to “showed”

L13. consider to change “lacking” to “limited”

Figure 2

Check “Transportres”

Check fonts in Tables 1 and 2, Figures 1 and 2, L41-43

REFERENCES

L28: I would suggest two recent references on the biosynthetic pathway of melatonin:

Molecules 2017, 22, 272; doi:10.3390/molecules22020272 Molecules 2018, 23, 1164; DOI: 10.3390/molecules23051164

Author Response

Rev.1

We have deleted the final term of the title.

The changes suggested in the Abstract have been incorporated, and some sentences have been rewritten.  A definition of ROS and the names of elements have been included, among others corrections.

New Keywords have been incorporated, as suggested.

All the minor changes suggested and some new references have been incorporated in the new version. Also, some paragraphs have been rewritten. The mistakes have been corrected.

On page 2, L5, the text has been reordered according to the reviewer’s suggestion.

On page 3 L28 (section 2), the new paragraph suggested and two refs. have been incorporated.

On Figure 2, the error in the word TRANSPORTERS has been corrected.

Reviewer 2 Report

General comments:

Considering current problems in environmental science, phytoremediation is a really an important issue. This review paper is focusing on the role of melatonin in an enhancement of the phytoremediation. The paper is interesting and might be considered for publication in the “Applied Sciences” after major revision. Major revision is needed to improve structuring and clarity of the information. Also the intensive English editing is needed. Some specific comments are given below. The authors should dedicate much more efforts to the manuscript editing. Some sentences are very long and hardly can be understood. The abstract part should be rewritten to make the main messages clearer and more concise.

Specific comments:

Lines 8-10: The sentence is long and hard to understand. I am asking to revise it.

Line 15:  ”especially in environmental stress conditions”, did you mean “under environmental stress conditions?”

Lines 14-18: Again, very long and complicated sentence

Line 19: The abbreviation ROS is given for the first time here without an explanation, it is not clear what does it mean

Lines 23-25: the chemical elements’ name should be first given in full here, e.g. sodium (Na), phosphorus (P) e.t.c

Line 25: … seems to be a useful option

Lines 31-33: does the number of the key words is matching to the requirements of the journal?

Line 42: when chemical elements are described for the first time, their full name should be given (see my previous comment to the abstract)

What happened to the line numbering after line 43?

Line 50: The first sentence in this paragraph is incomplete, the meaning is not clear

Line 11: “Brassicaceae species are known to have 10 exceptionally high metal accumulating capacities.” – reference is needed here

Lines 18-23: the sentence is very long and complicated

Line 24: again, the abbreviation is given without an explanation

Line 27: subheading needs revision, as the word “collaborate” can’t be used here

Line 35: references are needed here

Line 36: reference is needed at the end of the sentence

Lines 42-50: an extremely long sentence, should be rephrased

In the page 4, line numbering is starting again from the beginning, line numbering should be consistent through the entire manuscript

Page 4, line 23: start new sentence after the reference

Line 31: the year is needed Tan et al. (?)

Page 5, lines 5-6: incomplete and unclear sentence

line 16:  “combination of Cd with selenium or zinc” use either elements with their full names, or the letters in ne sentence

line 20: reference is needed

Table 1 is good and informative!

After page 7 the page numbering is missing

next line 20: it is not possible to use “in a recent paper”, use “recent study” or the other re-wording.

Then the other pager numbering system!

page 2 of 22, line 50: reference is missing

Table 2 is fine

Figure is fine

Author Response

Rev.2

Some sentences have been shortened. The Abstract has not been substantially modified since it seems to us that it provides the necessary and sufficient information about the manuscript. The last sentences points to the most interesting contributions that melatonin can make to phytoremediation.

The paragraph in Abstract L8-10 has been changed, in accordance with Reviewer 1.

All the names of chemical elements have been revised, and given in full the first time.

Some new refs. have been incorporated.

The sentences of P3 L18-23 and L42-50 have been rewritten.

We are happy that the reviewer liked Tables and Figures. Much appreciated.

Reviewer 3 Report

Abstract - lines 7-10 - these are obvious information. I would just delete them.

Introduction - lines 20-22 - instead of text please provide a decent table with some examples - would be beneficial for the readers

Plants are important but in fact the mutual dependence between plants and bacteria/fungi is crucial for all remediation processes - thus instead of another introduction on benefits of phytoremediation I would go into the details

Lines 28-32 - references are needed

Line 37 - one reference instead of several

table 1 - add possible explanation, summarize the observations

table 2 - says nothing - add concentrations, do not mix chemical and physical / biological factors in one chart - it is not easy to follow authors' idea behind this table

After relatively nice introduction and decent state of the art, the summary is missing. No future considerations are given. Authors should work on this mini-review in order to provide a well cited manuscript. Now it is just a brief description of available data. 

Author Response

Rev. 3

The reviewer suggests adding one more Table with the hyperaccumulatory species mentioned. We believe that with two tables already, one of them very extensive (Table 1) we should not include one more table.

In several points of the manuscript reference is made to the importance in the remediation processes of the use of microorganisms.

Some refs. have been incorporated.

The reviewer suggests giving more information in Table 1 and extending Table 2. In the text, the information on the items of each of the tables is considerably expanded. In the case of Table 2 it is not possible to separate chemical, physical and biological factors since in such cases the studies involve all those stressors.

Finally, the reviewer suggests introducing a new section (Summary). We believe that section 5 perfectly fulfills that function. Also in this section, new perspectives applied to remediation with the use of melatonin are suggested. All this is compiled in Figure 2, which we believe provides many of the details suggested.

Round 2

Reviewer 2 Report

I have one comment now, line 9 (contribute to, but not in)

Reviewer 3 Report

If authors present no will to enhance scientific value of the manuscript and consider it a final version then I shall not bother to be involved in further evaluation of the manuscript.